# Antibiotic Exposure, Common Morbidities and Main Intestinal Microbial Groups in Very Preterm Neonates: A Pilot Study

**DOI:** 10.3390/antibiotics11020237

**Published:** 2022-02-12

**Authors:** Nicole Bozzi Cionci, Laura Lucaccioni, Elisa Pietrella, Monica Ficara, Caterina Spada, Paola Torelli, Luca Bedetti, Licia Lugli, Diana Di Gioia, Alberto Berardi

**Affiliations:** 1Department of Agricultural and Food Sciences, University of Bologna, 40127 Bologna, Italy; nicole.bozzicionci@unibo.it (N.B.C.); diana.digioia@unibo.it (D.D.G.); 2Pediatric Unit, Department of Medical and Surgical Sciences for Mothers, Children and Adults, University of Modena and Reggio Emilia, 41124 Modena, Italy; 3Pediatric Unit, Department of Medical for Mothers and Children, Ramazzini Hospital, 41012 Carpi, Italy; e.pietrella@ausl.mo.it; 4Pediatric Unit, Department of Medical for Mothers and Children, Bufalini Hospital, 47521 Cesena, Italy; monicaficara@gmail.com; 5Neonatal Intensive Care Unit, Department of Medical for Mothers and Children, Bufalini Hospital, 47521 Cesena, Italy; catespa@gmail.com; 6Neonatal Intensive Care Unit, Department of Medical and Surgical Sciences for Mothers, Children and Adults, University of Modena and Reggio Emilia, 41124 Modena, Italy; torelli.paola@aou.mo.it (P.T.); luca.bedetti@unimore.it (L.B.); lugli.licia@aou.mo.it (L.L.); alberto.berardi@unimore.it (A.B.); 7PhD Program in Clinical and Experimental Medicine, University of Modena and Reggio Emilia, 41124 Modena, Italy

**Keywords:** neonates, preterm, gut microbiota, antibiotic exposure

## Abstract

Prematurity exposes newborns to increased risks of infections and it is associated with critical morbidities. Preterm infants often require antibiotic therapies that can affect the correct establishment of gut microbiota. The aim of this study was to investigate targeted intestinal bacteria in preterm neonates with common morbidities and receiving antibiotic treatments of variable duration. Stool samples were collected after birth, at 15, 30 and 90 days of life. qPCR quantification of selected microbial groups (*Bifidobacterium* spp., *Bacteroides fragilis* group, Enterobacteriaceae, *Clostridium* cluster I and total bacteria) was performed and correlation between their levels, the duration of antibiotic treatment and different clinical conditions was studied. An increasing trend over time was observed for all microbial groups, especially for *Bifdobacterium* spp. Prolonged exposure to antibiotics in the first weeks of life affected *Clostridium* and *B. fragilis* levels, but these changes no longer persisted at 90 days of life. Variations of bacterial counts were associated with the length of hospital stay, feeding and mechanical ventilation. Late-onset sepsis and patent ductus arteriosus reduced the counts of *Bifidobacterium*, whereas *B. fragilis* was influenced by compromised respiratory conditions. This study can be a start point for the identification of microbial biomarkers associated with some common morbidities and tailored strategies for a healthy microbial development.

## 1. Introduction

Colonization of the neonatal gut microbiota begins in utero [1,2,3,4] and its development is influenced by several factors, such as the lower gestational age, prolonged hospital stay, mode of delivery, maternal microbiota, use of antimicrobials and the enteral feeding [5].

Compared with healthy full-term infants, preterm infants have a lower bacterial diversity of the gut microbiota, with a prevalence of potential pathogenic bacteria [6] and lower amounts of beneficial bacteria [7]. The gut colonization sequence of healthy full-term infants, with the initial predominance of facultative anaerobes followed by strict anaerobes, may be compromised in preterm neonates. Gibson et al. [6] demonstrated that the gut microbiota of preterm infants aged one to two months hosts a higher abundance of Enterobacteriaceae and Staphylococcaceae, which are facultative anaerobes, a lower abundance of Clostridiaceae and Bifidobacteriaceae, which are strict anaerobes, and a lower abundance of Streptococcaceae and Lachnospiraceae.

In addition, the gastric pH of preterm neonates is higher compared with full-term ones, leading to a higher risk of bacterial infections [8]. Finally, immature natural barriers and insufficient immune responses and the many invasive devices used to provide life-supporting care increase the susceptibility of preterm neonates to bacterial infections.

Unfortunately, antibiotic exposure is frequent in preterm infants, and it is associated with reduced richness of species [6], having potential short- and long-term consequences on the correct development of the microbial structure and immune system [9,10,11]. An abnormal intestinal microbiome influences the host immunity [12,13,14], increasing the risks of necrotizing enterocolitis (NEC) [15,16,17,18] and late-onset sepsis (LOS) [19]. In preterm neonates, a prolonged course (>five days) of empirical antibiotics at birth may lead to a lower diversity of intestinal microbiota [20].

The current study aims to evaluate changes in selected microbial groups in the gut of extremely preterm or extremely low birth weight neonates aged less than 90 days, after different durations of antibiotic treatments and/or in the presence of common morbidities of preterm neonates. We also investigated the association between these morbidities and gut bacterial counts, to identify bacterial markers potentially predictive of the main morbidities of preterm neonates.

## 2. Results

Twenty-three preterm infants were enrolled, seventeen of whom completed the T0 to T3 follow-up; as for the remaining six infants, three were lost between T1 and T2 (one died and two were given probiotics) and three were lost between T2 and T3 (two died and one was given probiotics).

Table 1 shows demographics, clinical characteristics, length of stay, and antibiotic exposure of infants enrolled in the study. All but three neonates were given antibiotics at birth.

The characteristics regarding the main morbidities and durations of antibiotic treatments for each neonate are reported in Figure 1. 

Figure 2 shows the mean counts of the microbial groups analyzed by quantitative PCR (qPCR). A general increasing trend over time was observed for all microbial groups and total bacteria. *Bifidobacterium* spp. significantly increased at T3 with respect to T0, T1 and T2, the *B. fragilis* group significantly increased at T3 with respect to T0, Enterobacteriaceae and total bacteria were significantly lower at T0 with respect to the other timepoints and *Clostridium* cluster I was significantly higher at T3 with respect to T0 and T1.

Table 2 shows the variables associated with changes in microbial groups. Days on total parenteral nutrition were inversely associated with *Bifidobacterium* spp. at T2, whereas days at full enteral feeding were inversely associated with the amount of *Bifidobacterium* spp. at T3. Days at the beginning of enteral feeding were not associated with counts of microbial groups.

No associations were found between antibiotic therapy (at T0 and T1) and microbial groups. However, a borderline association was found between *Clostridium* cluster I at T1 and total days on antibiotics (from 0 to 30 days of life). Total days on antibiotics (from 0 to 90 days of life), were also significantly associated with Clostridium cluster I (at T1) and inversely associated with *B. fragilis* group counts at T2.

The association between potential predictors, including days of hospital stay, days on mechanical ventilation, days on parenteral nutrition, days at full enteral feeding, days at the beginning of enteral feeding, total days on antibiotics (from 0 to 15 – from 0 to 30 and from 0 to 90 days of life) and bacterial counts at each time of the study was evaluated using a multiple linear regression model (Appendix A Appendix A). Although some statistical significances were evidenced for *Clostridium* cluster I at and the *B. fragilis* group, the small sample size did not allow us to draw firm conclusions. 

Counts of *Bifidobacterium* spp. were significantly lower in males with respect to females. Table 3 presents changes in microbial groups at different time points according to morbidities. Changes in selected bacterial groups were associated with a full course of prenatal steroids, neonatal birth weight, and morbidities of premature newborns. Compared with neonates with adequate birth weight, neonates small for gestational age (SGA) showed significantly lower amounts of *B. fragilis* group at T0. Furthermore, an increase in the *B. fragilis* group was evidenced at T1 and T2 in infants who received a full prenatal steroid course as compared with those receiving inadequate or no steroid course. Infants with LOS had lower levels of *Bifidobacterium* spp. at T2. PDA (patent ductus arteriosus) was associated with fluctuations of counts of microbial groups. Lower amounts of *Clostridium* cluster I were observed at T0 and *Bifidobacterium* spp. at T2; high amounts of the *B. fragilis* group were observed at T3 in infants with PDA. Moreover, infants with NEC showed lower counts of *Clostridium* cluster I at T0 and infants with retinopathy of premature (ROP) had a borderline increase in the *B. fragilis* group at T3. Counts of *B. fragilis* group increased in infants with BPD. Intrapartum antibiotic prophylaxis, membrane rupture lasting more than 18 h and delivery with intact membranes did not show significant differences among microbial groups.

## 3. Discussion

The infant gut microbiota is significantly affected by a number of perinatal factors [21,22]. Changes in the main intestinal microbial groups in very preterm neonates during the first weeks of life were analyzed, according to antimicrobials administered, days of treatment and the presence of common morbidities of preterm neonates.

As expected, counts of *Bifidobacterium* were generally lower in the first 30 days of life compared with mean values of healthy full-term infants available in the literature [21,22,23,24,25,26,27,28,29,30]. However, counts reached normal values (higher than 7 Log CFU/g of faeces) at age three months. In addition, counts of the *B. fragilis* group were particularly low in preterm as compared with full-term newborns reported in the literature [31,32]. In preterm neonates enrolled in this study, the qPCR analyses showed increased amounts of total bacteria and bacterial groups from birth to 90 days of age. Indeed, the gut microbiota develops rapidly after birth, when infants have contact with a rich and diverse microbial environment. Infants are rapidly colonized by microorganisms that originate primarily from the maternal microbiome, in particular from the genital tract and feces [22,33]. The increase in bacterial counts found in the current study can be attributed to the persistent neonatal exposure to microorganisms originating from the NICU environment or from feeding; this exposure leads to an increase in the gut bacterial members and, potentially, to the microbial diversity [34]. The increase in Enterobacteriaceae over time is consistent with a previous study [6], demonstrating a higher abundance of facultative anaerobes (such as Enterobacteriaceae) in preterm infants aged less than two months. This is a critical period for the health of preterm neonates. Indeed, multidrug-resistant pathogens belonging to the family of Enterobacteriaceae, which are commonly associated with nosocomial infections, are abundant in preterm neonates [6]. At a later stage of life (90 days), a prominent increase in *Bifidobacterium* spp. and *Clostridium* cluster I, which are strictly anaerobic bacteria, was also evidenced; this finding is consistent with previous studies in preterm infants [6]. Furthermore, changes in bacterial populations were associated with several variables, such as prolonged length of hospital stay, mode of feeding, mechanical ventilation and longer duration of antibiotic treatments. Prolonged empirical antibiotics at birth affect the initial establishment of intestinal microbiota in preterm neonates, increasing the risks of complications, such as NEC and LOS [35,36,37,38,39,40]. Therefore, it is possible to speculate that a prolonged antibiotic exposure may affect the composition of the microbiome, being associated with *Clostridium* cluster I and inversely associated with the *B. fragilis* group. These data suggest that a prolonged antibiotic exposure, although often necessary, can lead to a potential imbalance of composition of the gut microbiome. However, the current study failed to demonstrate an association between selected microbial groups and intrapartum antibiotic prophylaxis, which is known to reduce the bifidobacterial population in full-term neonates [21]; however, most neonates in this study were still treated with antibiotics in the first weeks of life.

The mode of feeding plays an important role for the gastrointestinal development. Enteral nutrition promotes the development of both mucosal immune system and gut microbiota, thereby preventing infections in preterm neonates [41]. Therefore, prolonged total parenteral nutrition may reduce Bifidobacteria. Furthermore, selected bacterial groups were associated as early as T0 with gender, antenatal steroid treatment and common morbidities of preterm neonates (LOS, NEC, PDA, SGA). The exact mechanisms underlying these associations are sometimes unclear, although an increase in the *B. fragilis* group at T0 in neonates born as SGA might depend on their higher rates of caesarean section.

Many microorganisms responsible for LOS in preterm neonates may originate from the gastro-intestinal tract and translocation of gut microbes may account for most cases; decreased bacterial diversity and predominance of Gram-positive cocci, such as staphylococci, in early fecal specimens have been both associated with the development of LOS [19]. Alterations in the fecal microbiota frequently occur prior to LOS, depending on timing and site [42]. Indeed, an abnormal microbiota containing LOS potential pathogens prior to or concurrent with the onset of LOS has been demonstrated [2,43]. The neonatal gut is especially prone to colonization with aerobic Gram-negative bacilli, to the detriment of beneficial bacteria, such as bifidobacteria and lactobacilli, that may later predispose the neonates to septicemia [7]. In the current study, LOS was associated with lower levels of bifidobacteria at T2, a finding that is consistent with previous studies [44]. It is possible that the reduction in these beneficial bacteria may have favored the development of other enteric pathogens responsible for LOS, not specifically sought by qPCR.

Perturbations of the intestinal microbiome and raised inflammatory responses have been implicated in the development of NEC in preterm infants [45,46,47]. NEC has been associated with a variety of pathogens, since there is not a single microorganism responsible for NEC [2]. Furthermore, the age at the onset of NEC may affect intestinal microbial profiles. Increased levels of Firmicutes and decreased amounts of Gammaproteobacteria have been detected when NEC occurs prior to ten days of life. In contrast, the onset of NEC beyond 10 days of life has been associated with decreased levels of Firmicutes (especially Negativicutes) and increased amounts of Gammaproteobacteria [48]. Epidemiological studies and animal models support the involvement of anaerobic bacteria, particularly those belonging to *Clostridium* genus, in the development of NEC although their role in NEC is still unclear. Colonization by clostridia seems harmful, [49] although lower amounts of clostridia in preterm neonates with NEC have been reported; in such cases they would prevent the inflammatory response associated with NEC [50]. Low amounts of *Clostridium* cluster I at birth in the two neonates with NEC were found. Therefore, it is possible that within the large *Clostridium* genus different microbial species or strains act in opposite ways, by favoring or preventing NEC.

*Clostridium* cluster I was reduced at birth also in infants who subsequently were diagnosed with PDA; this finding would suggest the role of *Clostridium* cluster I in the already proven association between PDA and NEC [51]. To our knowledge, the PDA of preterm neonates has not been previously associated with an abnormal gut microbiota; PDA may reduce blood intestinal perfusion, delaying the achievement of a full enteral feeding, and changing intestinal flora by increasing the amounts of potentially harmful bacteria (i.e., Enterobacteriaceae and *B. fragilis* group), and decreasing beneficial ones (such as bifidobacteria, as evidenced in our study).

Fluctuations of some microbial groups (especially *B. fragilis*) in neonates with bronchopulmonary dysplasia were also showed. Both the gut and the lungs, besides sharing some anatomical epithelial similarities, can promote immune responses. Gut bacteria can reach lungs after gastroesophageal reflux, inhalation or vomit. The damage to the integrity of the epithelium may enable bacteria or their components and metabolites to enter the circulatory system (causing systemic inflammation), and to reach the respiratory tract through the lymphocytes migration [52]. Moreover, exposure to an oxygen-rich environment (such as during mechanical ventilation) might impair the development of the nascent gut microbiome, that is physiologically dominated by anaerobic bacteria which affect its immunological properties [53,54]. Previous studies on airway microbiomes [55,56], did not find associations between *Bacteroides* and BPD. Some investigators [57] suggested that an abnormal microbiome may affect the risk of BPD, asthma and allergic diseases later in life. We found higher amounts of *B. fragilis* in respiratory disorders (i.e., BPD, prolonged mechanical ventilation). The increased amounts of *B. fragilis* may reflect an abnormal composition of the gut microbiome that can have consequences for the health status.

Finally, by using a murine model, investigators [58] found that antenatal steroids may affect the epigenome of the host and early colonization of microbiome, leading to increased levels of strict anaerobic bacteria, such as clostridia. Higher levels of bifidobacteria after antenatal steroid treatment were found. Bifidobacteria are strict anaerobic microorganisms; antenatal steroids would have therefore a beneficial effect on the microbiome, since usually bifidobacteria are less abundant in premature neonates [6,35,59].

This study has some important limitations. Firstly, the samples size of infants in the study is small, and some neonates did not complete the entire fecal sample collection. Furthermore, neonates were very preterm; consequently, they often had overlapping morbidities, complicating their clinical course and prolonging their hospital admission. These complications make it difficult to establish with certainty the relevance of some single associations, despite some variables were statistically significant. Previous investigators have sometimes addressed these major problems by comparing early preterm neonates with infants of higher gestational age and birth weight. However, this comparison would not be fully appropriate since neonates with higher gestational age and birth weight have much less exposure to antibiotics and their clinical course is much less complicated.

## 4. Materials and Methods

### 4.1. Study Design and Participants

We conducted a single center, prospective, observational cohort study (from 1 March 2017 to 31 August 2018) concerning preterm infants admitted to our NICU. Infants under 28 weeks’ gestation and/or a birth weight under 1000 g were enrolled. The local ethics committee approved the project (Prot N 192/16) and written informed consent was obtained from parents of neonates included in the study. Fecal samples were longitudinally collected within the first 48 h of life (T0), at 15 (T1), 30 (T2) and 90 (T3) days of life.

Antibiotic exposure was assessed as the number of neonates receiving antibiotics from 0 to 3 days of life. Because most neonates (20 out of 23) were given antibiotics in the first 3 days of life, we also calculated the median days of antibiotic exposure in three different periods of life (from 0 to 15, 30 and 90 days of life, respectively). According to our local protocol, ampicillin plus gentamicin (from birth to three days of life) and oxacillin plus amikacin (beyond three days of life) were given as empirical antibiotics in most cases, pending cultures and laboratory results. Clinical data regarding length of stay in NICU, common morbidities, parenteral infusions and enteral feeding were collected (Table 1).

Exclusion criteria were major malformations and/or the administration of probiotics.

### 4.2. DNA Extraction from Fecal Samples

Meconium and fecal samples were collected from neonates and subsequently stored at − 80°C. Samples were subject to DNA extraction using QIAamp DNA Stool Mini Kit (Qiagen, West Sussex, UK) with a slight modification of the standard protocol: a supplementary incubation at 95 °C for five minutes of the stool sample with the lysis buffer was added to enhance the bacterial cell rupture [21]. The purity of the DNA was determined by measuring the ratio of the absorbance at 260 and 280 nm (Infinite R 200 PRO Nano Quant, Tecan, Mannedorf, Switzerland) and the concentration was evaluated by Qubit R 3.0 Fluorometer (Invitrogen, Life Technologies, CA, USA).

### 4.3. Absolute Quantification of Selected Microbial Groups Using Quantitative PCR (qPCR)

A qPCR analysis was performed on DNA extracted from fecal samples by using the Fast SYBR^®^Green Master Mix (Applied Biosystems, Foster City, USA) assay and optimized concentrations of primers [23,24]. We quantified total bacteria and selected microbial groups, which are usually investigated in studies targeted to infants [25,26], i.e., *Bidobacterium* spp., *Bacteroides fragilis* group (which includes the most abundant species in the human gut: *B. fragilis*, *B. distasonis*, *B. ovatus*, *B. thetaiotaomicron*, *B. vulgatus*), *Clostridium* cluster I and Enterobacteriaceae. For each target microorganism, standard curves were created by using 16S rRNA PCR product of type strains [23,27]. Data were then converted to obtain the number of microorganisms as Log CFU/g feces, according to the rRNA copy number [26]. For total bacteria, the average of the 16S rRNA genes calculated on 10,996 records for bacteria according to rrDB was used as the rRNA copy number [60].

### 4.4. Statistical Analysis

Data are expressed as median ± interquartile and percentages. Nonparametric statistical analysis was performed by using SPSS (version 20). Chi-squared tests were used to compare discrete variables. Inter-group comparison for continuous variables was performed with the Mann–Whitney U test and Kruskal–Wallis when appropriate; for the first test, the Z test statistic and Asymptotic Significance were corrected for ties and for the second one, Bonferroni correction was used. Correlation between variables was determined by using the Spearman test. The association between potential predictors and bacterial counts was evaluated using a multiple linear regression model; beta has been reported as standardized coefficient and standard error as unstandardized coefficient. Statistical significance was set at *p* value ≤ 0.05.

## 5. Conclusions

This study addresses a population at high risk of developing an abnormal gut microbiota, such as infants of a lower birth weight and gestational age. Although the underlying mechanisms are not entirely clear, many common morbidities of preterm neonates may affect counts of both total bacteria and targeted microbial groups from birth to 90 days of age. Prolonged exposure to antibiotics in the first weeks of life affects the intestinal microbiota, although differences seem to no longer persist at the age of 90 days. Further studies in larger cohorts would assess more accurately how common morbidities affect changes in the intestinal microbiome of very preterm neonates.

## Figures and Tables

**Figure 1 antibiotics-11-00237-f001:**
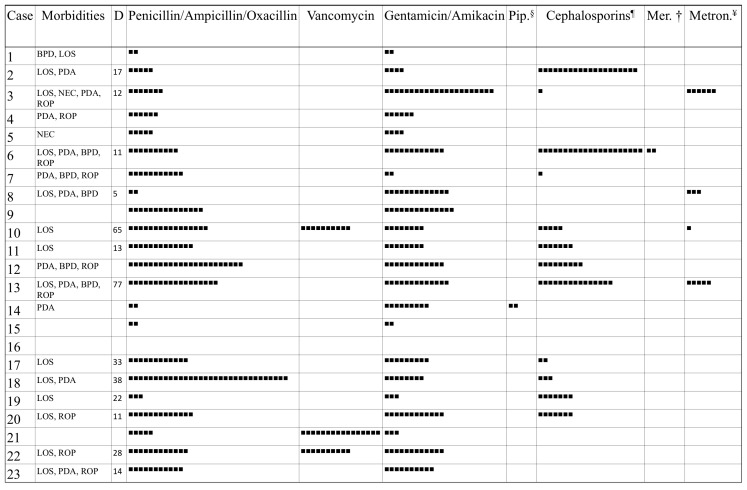
Morbidities of the 23 preterm neonates and antibiotics administered. BPD, Bronchopulmonary Dysplasia; LOS, Late-Onset Sepsis; NEC, Necrotizing Enterocolitis; PDA, Patent Ductus Arteriosus; ROP, Retinopathy of Prematurity. Each square (■) corresponds to one day of antibiotic therapy. D, Days of life at onset of LOS; § Piperacillin-Tazobactam; ¶ Cephalosporins of second and third generation; † Meropenem; ¥ Metronidazole.

**Figure 2 antibiotics-11-00237-f002:**
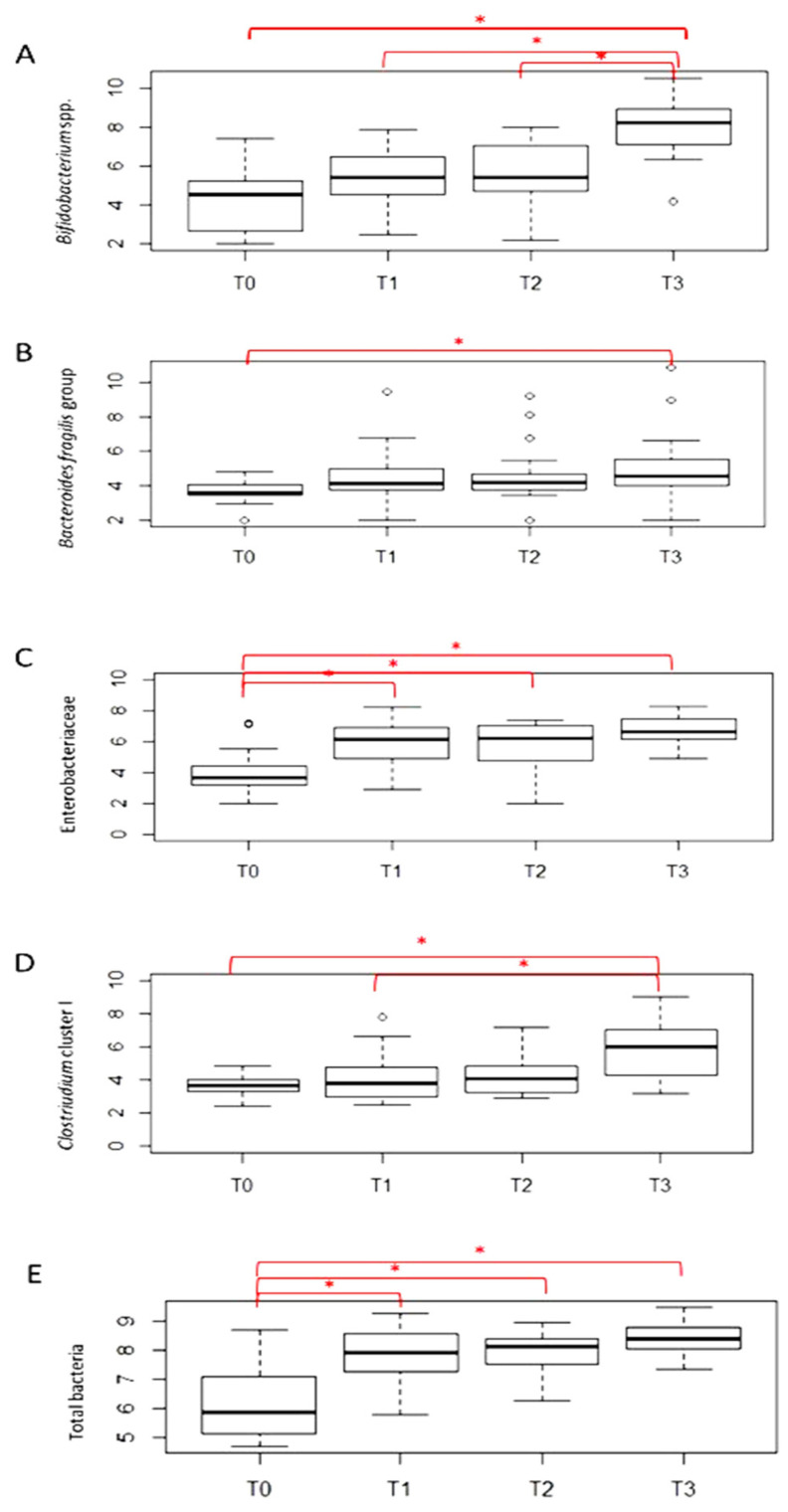
qPCR counts of selected microbial groups. Counts are expressed as Log_10_CFU/g feces; * indicates *p* ≤ 0.05.

**Table 1 antibiotics-11-00237-t001:** Characteristics of the study population. Data are presented as median (IQR) or *n* (%).

Variables	Cases, *n* = 23
Gestational age, wks	26 (25.6–27.4)
Birth weight, g	835 (740–970)
PPROM ≥ 18 h *	9 (39.1%)
Vaginal delivery	5 (21.7%)
Cesarean section with rupture of membranes or during labor	12 (52.2%)
Caesarean section with intact membranes and without labor	6 (26.1%)
SGA	3 (13%)
Male sex	12 (52.2%)
Twins	10 (43.5%)
Antenatal steroids full course incomplete course	20 (87%) 17 (74%) 3 (13%)
IAP	9 (39.1%)
Apgar score at 5th minute	7 (3–8)
Days at the beginning of enteral feeding	2 (1–2)
Days at full enteral feeding	34.5 (25.5–40.3)
Total days on parenteral nutrition	26 (20–36)
MV	18 (78.3%)
Days on MV	7 (0–18)
PDA (medical treatment) §	11 (47.8%)
LOS °	13 (56.6%)
NEC (Bell stage ≥ 2)	2 (8.7%)
BPD †	7 (33.3%)
ROP	9 (39.1%)
Length of hospital stay, days	73 (60–107)
Antibiotic exposure (0–3 days of life)	20 (87.0%)
Total days on antibiotics (0–15 days of life)	6 (2–10)
Total days on antibiotics (0–30 days of life)	10 (5–16)
Total days on antibiotics (0–90 days of life)	13 (7–22)

BPD = bronchopulmonary dysplasia; IAP = intrapartum antibiotic prophylaxis; LOS = late-onset sepsis; MV = mechanical ventilation; NEC =necrotizing enterocolitis; PDA = patent ductus arterious; PPROM = Preterm premature rupture of membranes; ROP = Retinopathy of prematurity.* duration of membrane rupture was unknown in 2 cases. † Oxygen support at 36 weeks postmenstrual age; 2 neonates who died before 36 weeks were excluded from calculation. § 2 cases underwent surgical treatment. ° LOS was due to group B *Streptococcus* (*n* = 4), *Escherichia coli* (*n* = 3), *Klebsiella pneumoniae* (*n* = 2), *Enterobacter aerogenes* (*n* = 1), *Proteus mirabilis* (*n* = 1), *Staphylococcus aureus* (*n* = 1), *Staphylococcus epidermidis* (*n* = 1).

**Table 2 antibiotics-11-00237-t002:** Changes in microbial groups according to some variables of the study population.

Variables	T0 (Within 48 h of Life)	T1 (15 Days of Life)	T2 (30 Days of Life)	T3 (90 Days of Life)
Days of hospital stay	NS	*Bifidobacterium* spp. (*p* = 0.054; r −0.43) *B. fragilis* group (*p* = 0.008; r −0.56)	*B. fragilis* group (*p* = 0.006 r −0.58)	*B. fragilis* group (*p* = 0.03; r 0.54)
Days on mechanical ventilation	NS	NS	*Bifidobacterium* spp. (*p* = 0.037; r −0.46)	*Bifidobacterium* spp. (0.039; r −0.52) *B. fragilis* group (*p* = 0.018; r 0.58)
Days on parenteral nutrition	NS	NS	*Bifidobacterium* spp. (*p* = 0.013; r −0.53)	NS
Days at full enteral feeding	NS	NS	NS	Bifidobacterium spp. (*p* = 0.015 r −0.6)
Days at the beginning of enteral feeding	NS	NS	NS	NS
Total days on antibiotics (from 0 to 15 days of life)	NS	NS	NS	NS
Total days on antibiotics (from 0 to 30 days of life)	NS	*Clostridium* cluster I (*p* = 0.053; r 0.43)	NS	NS
Total days on antibiotics (from 0 to 90 days of life)	NS	*Clostridium* cluster I (*p* = 0.03; r 0.47)	*B. fragilis* group (*p* = 0.02; r −0.52)	NS

NS, not significant.

**Table 3 antibiotics-11-00237-t003:** Changes in microbial groups at different time point according to morbidities of preterm neonates.

	T0	T1	T2	T3
SGA	*↓B. fragilis* group (*p* = 0.047)	NS	NS	NS
Antenatal steroids	NS	*↑B. fragilis* group (*p* = 0.036)	*↑B. fragilis* group (*p* = 0.011)	NS
LOS	NS	NS	*↓Bifidobacterium* spp. (*p* = 0.018)	NS
PDA	*↓Clostridium* cluster I (*p* = 0.036)	↑Enterobacteriacee (*p* = 0.06)	*↓Bifidobacterium* spp. (*p* = 0.029)	*↑B. fragilis* group (*p* = 0.023)
NEC	*↓Clostridium* cluster I (*p* = 0.029)	NS	NS	NS
ROP	NS	NS	NS	*↑B. fragilis* group (*p* = 0.056)
BPD	NS	NS	NS	*↑B. fragilis* group (*p* = 0.013)

BPD, bronchopulmonary dysplasia; IAP, *intrapartum* antibiotic prophylaxis; LOS, late-onset sepsis; NEC, necrotizing enterocolitis; PDA, *patent ductus arteriosus*; ROP, retinopathy of premature; SGA, small for gestational age. NS: not significant.

## Data Availability

De-identified individual participant data presented in this study are available on request from the corresponding author. The data are not publicly available due to the need for use in further research.

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
