# Peer review of "Antibiotic Exposure, Common Morbidities and Main Intestinal Microbial Groups in Very Preterm Neonates: A Pilot Study"

_antibiotics, 2022, doi:10.3390/antibiotics11020237_

Round 1

Reviewer 1 Report

Thank you for submitting this interesting manuscript to Antibiotics. Overall, this is a very well written manuscript. However, here are some suggestions in case you consider taking them into account with the hope that they will be useful to improve the quality of it.

Major comments.

The sentence (lines 189-191) "We also found that prolonged total parenteral nutrition impaired gut colonization with bifidobacterial, suggesting a detrimental effect on the normal development of the intestinal microbiota" perhaps should be qualified since it is possible that the alteration of the microbiota is not due directly to the use of prolonged parenteral nutrition, but to the fact that the patients who need it is because they are not receiving (cannot receive) enteral feeding.

Likewise, some associations between common morbidities of prematurity, such as PDA, and alterations of the microbiota, especially in T0, would need more discussion regarding biological plausibility. Various confounding factors may intervene, which in a deeper multivariate analysis could make this association disappear.

Lines 207-209. Given that 13 (56.6%) patients presented late sepsis (LOS), it would be interesting to know the "time sequence" regarding the decrease in Bifidobacterium in T2 (30 days). If LOS occurred earlier, it would be difficult to attribute their origin to this phenomenon, as explained previously in the same paragraph.

Line 223: It says “We found low amounts of Clostridium cluster I at birth in the 3 neonates with NEC.” But in Table 1, where the “Characteristics of the study population” are shown, there are only 2 patients with NEC.

Minor comments.

A call for Table 2 seems to be lacking in the text, probably between lines 109 and 117.

Please, consider explaining the abbreviation “qPCR” the first time it appears.

Typos.

Line 21: “…increased risks of infections and is associated with of critical morbidities.”

Reviewer 2 Report

Review Cionci et al. Antibiotic exposure, common morbidities and main microbial groups in very preterm neonates: a pilot study

Many thanks for the opportunity to review the work of Cionci and colleagues

In this manuscript, the authors describe the association between the clinical condition including antibiotic treatment of preterm neonates and their respective microbiota.

The paper is well written. In the following I will make some suggestions that will hopefully help to improve the manuscript in order of appearance.

My two main concerns pertain to:

  1. Report antibacterial treatments more detailed
  2. Develop a regression model (see details below).

Abstract:

  • Line 21: “is associated with of critical”: please delete “of”.
  • Line 27: please change “Enterobacteriaceae” to “Enterobacterales” throughout the manuscript.
  • Line 37: please consider reframing “correct microbial development”, which suggests one correct way which should be aimed for.

Introduction:

  • Lines 54-57: maybe consider adding immaturity of natural barriers as another factor contributing to a higher susceptibility for infections.

Results:

  • In general: I would like to see the antibiotic treatment choices and potential differences per treatment regimen, since treatment of neonatal sepsis can be rather narrow (e.g., Ampicillin +/- Gentamicin), or very broad (e.g., a carbapenem), and although the authors explain in the methods that mostly either ampi+genta or oxa+amikacin were used, I would be curious to see if some patients, maybe even alater time point, had “broader treatments”, and if this had an impact on the microbial counts.

Discussion:

  • There is a lack of a control group, e.g., term neonates without antibiotic treatment. Please discuss this more elaborately.

Materials and Methods:

  • I would suggest to perform a regression analysis with the bacterial count (overall and subgroups also) as the dependent variable, to better model the potential association with the described variables, which most probably have large interactions between each other.

Conclusions:

  • No comments.

Reviewer 3 Report

This manuscript by Nicole Bozzi Cionci et al. describe the association between microbial perturbations in very preterm neonates.

This manuscript is simple, easy-to-read, but deserve minor revisions.

Global : prefer passive form. Number below 12 must be written in full letters. italicize "i.e.", "et al.", bacterial names, 

Table 1 : do not use abbreviation , as that harden the understanding of the manuscript.

Results : do not precise statistical trends as they can be misleading.

Statistics : How did the authors considered multiple testing correction ? Some results with p value near to 0.05 could be impacted.

Methods : How was determined the number of patient to include (as some results are trends, it is crucial to explain)? Why have the authors not consider 16S DNA sequencing for example ? This is more expensive but lot more informative than just PCR. 

Round 2

Reviewer 2 Report

Many thanks to the authors for their hard work.

However, I still do not seem to understand how the authors conducted regression analyses, and what exactly they did (ANOVA? this is not regression). The Table in the supplements mirrors basically Table 2 from the main paper, and the only numbers that are given seem to be p-values and correlation coefficients. For a proper regression however, I would expect something more like incidence rate ratios, odds ratios, etc..

Please revise.

Author Response

Please, see the uploaded file.
